# Assessment of the Functional Level of Independence in Individuals with Mental Disabilities as Part of Special Education Diagnostics: Case Studies

**DOI:** 10.3390/ijerph192315474

**Published:** 2022-11-22

**Authors:** Michal Vostrý, Barbora Lanková, Ilona Pešatová, Lenka Müllerová, Helena Vomáčková

**Affiliations:** 1Research Centre, Faculty of Health Studies, Jan Evangelista Purkyně University in Ústí nad Labem, 400 96 Ústí nad Labem, Czech Republic; 2Department of Special and Social Education, Faculty of Education, Jan Evangelista Purkyně University in Ústí nad Labem, 400 96 Ústí nad Labem, Czech Republic; 3Department of Primary and Pre-Primary Education, Faculty of Education, Jan Evangelista Purkyně University in Ústí nad Labem, 400 96 Ústí nad Labem, Czech Republic

**Keywords:** mental functional diversity, functional degree of independence, special pedagogy, social adaptability

## Abstract

In this study we focus on the application of standardized tests aimed at evaluating the functional degree of independence in children (client 1, WeeFIM test; and client 2, FIM test) in special education diagnostics. The target group consisted of two clients with a diagnosis of mental functional diversity (*n* = 2; client 1: mild mental retardation, according to ICD-10: F70, aged 6.5 years; and client 2: moderate mental retardation, according to ICD-10: F71, aged 13.4 years). Special pedagogical intervention was primarily applied to the clients, focusing on identified deficits in the areas of cognitive, motor, and social skills. The presented results demonstrate the importance of the application of these tests in special pedagogy. An improvement in the observed indicators of the given tests was demonstrated for both probands after the intervention. The aim of this article was to draw attention to the suitability of using functional independence tests in special pedagogical practices. The authors discuss the further implications of this application for future practice.

## 1. Introduction

It is estimated that up to 1% of people in the general population have some degree of intellectual functional diversity. The diagnosis of intellectual functional diversity itself is based primarily on the clinical history, the level of intellectual abilities, and the level of adaptive functions. In addition to clinical evaluation, cognitive functions or adaptive functioning in ordinary daily activities are also diagnosed based on individually applied standardized tests [1,2]. The conceptual framework of intellectual functional diversity has evolved over the years primarily from the medical model. This model considers a person with an intellectual functional diversity who is limited in the ability to fulfil expected individual and social obligations. A functional limitation [3,4,5] is thus attributed to the individual. Advances in science and medicine are contributing significantly to the increasing number of individuals with serious illnesses. It is necessary to perceive this increase effectively and to be able to respond to it appropriately. Therefore, the attitudes of society or teaching staff are also important. It is these attitudes towards individuals with a certain type of functional diversity that determine future education and other approaches that are developed by the given individual and thus support social adaptability [5,6].

While the social model points to the fact that functional diversity in itself may not immediately indicate a certain degree of the intellectual functional diversity. Rather, the model points to a restrictive environment that limits the given individual to participate effectively in the expected individual or social duties. However, both models point to the same thing, the view of coping with activities of daily living, i.e., how dependent the individual is on care staff [7]. Intellectual functional diversity, formerly known as mental retardation, is a condition characterized by markedly below average intellectual functioning. It is accompanied by a disorder of adaptive behaviour that manifests itself before the age of 18. According to the ICD-10, we include mild, moderate, severe, and profound disabilities among the individual degrees. According to ICD F70–F79, mental retardation is a state of stopped or incomplete mental development, which is characterized in particular by a violation of skills, manifested during the developmental period, affecting all components of intelligence, that is, cognitive, speech, motor, and social abilities [8].

Intellectual abilities and social adaptability can change over time, and even reduced intellectual values can improve with exercise and rehabilitation. It is usually possible to measure mental retardation by standardized intelligence tests. Moderate mental retardation F71 IQ reaches values of 35 to 49 (which in adults corresponds to a mental age of 6 to 9 years). The result is a distinct developmental delay in childhood‚ though a great many children manage to develop to a degree of independence and self-sufficiency‚ achieving adequate communication and school skills. Adults will need varying degrees of support to work and to function in society [9,10].

In general, clinical signs of intellectual functional diversity are initially recognized during childhood. Predominant symptoms suggestive of intellectual functional diversity vary with age. For example, in infancy, if a severe intellectual deficit is present, there is a high probability that clinical symptoms will appear relatively early. Conversely, if a mild degree of the intellectual deficit is present, this functional diversity may not be fully recognizable at first. It may manifest itself later in childhood [11]. In the case of a mild intellectual functional diversity, difficulties are usually encountered in acquiring and understanding complex language and academic skills. In addition to language skills, individuals also have difficulty with arithmetic and writing skills. With appropriate support, they are able to write simple texts or fill out simple applications [12]. A moderately severe mental functional diversity has an impact on an individual’s rate of acquisition of basic language skills. This adoption is slow. Individuals show significant limitations in reading, writing or arithmetic. Likewise in other skills where understanding of basic concepts is required. Difficulties are also manifested in social communication, interpersonal interaction and understanding of social norms. Permanent support is often required to enable the development of meaningful family and personal relationships [4,13,14].

### 1.1. Social Adaptability

Mental functional diversity manifests itself in a lack of social, conceptual, and practical competencies. Social competency is demonstrated through interpersonal skills, social skills, self-esteem, trustworthiness, naivety, social problem solving, and the ability to follow social norms. Conceptual competency makes it possible to understand time, finance, or language. Practical competency involves the use of various tools to perform certain activities of daily life and also to communicate with other people. We acquire all these skills throughout our lives and we undertake them in response to common problems.

With age, these reactions become progressively more complex [15,16,17,18,19]. When there is appropriate support shown, an individual of mild intellectual functional diversity can reach a state where they acquire most of the competencies. These competencies are needed to manage normal daily activities. Individuals may also be gainfully employed as adults and be able to maintain independence in daily activities. However, as the degree of mental disabilities increases, this independence deteriorates [20]. Even in the case of moderately severe mental disabilities, individuals need considerable and permanent support, even in adulthood. With this support, they can be gainfully employed and as independent as possible in their daily activities. Individuals suffering from severe intellectual functional diversity require intensive support in all activities of daily life. This also includes hygiene and selfcare. These individuals are not able to make proper decisions or judgments that would affect their wellbeing. They require constant supervision and may present with maladaptive and self-harming behaviour which is a serious problem for the care of these individuals. The last type of functional diversity is profound mental functional diversity. Here, the condition is so severe that the individual is completely dependent on the care of others in all aspects of daily life. In some cases, individuals develop a certain capacity for nonverbal communication with the application of augmentative communication methods [4,21,22,23].

### 1.2. Selected Intervention Approaches

Nowadays, the emphasis is on individuals with mental disabilities to be able to live as full a life as possible. Participating in daily activities is also associated with this. It is the performance of daily activities that contributes to the internal and external view of a person as an autonomous and responsible being [24,25]. The rehabilitation field of occupational therapy plays a major role in the development of managing daily activities. It is primarily based on activities that the client considers meaningful and purposeful. Diagnostically, therapists thus use a top-down approach. First, they gather information about what is important to the client and based on that, the design of the intervention itself is developed. Standardized tests can be used to create a suitable goal and strategy for the development of daily activities. These tests, primarily used in the healthcare sector, determine the degree of dependence of the client on the nursing staff. Such tests include, for example, the Scales of Independent Behaviour, the Vineland Adaptive Behaviour Scales, the Barthel Index, as well as the FIM test for adults or the WeeFIM test for children under seven years of age [26,27]. The WeeFIM test is a tool used to indicate functional outcomes in children and is based on the FIM test, a widespread tool in adult rehabilitation facilities. The WeeFIM test contains 18 items and can be administered within 30 min. The aim of this tool is to measure changes in function and assess the burden of care over time. The WeeFIM test tool has excellent consistency and the scores provided are stable. Both mentioned tests are subject to a license [28,29,30].

Individuals with intellectual disabilities are an extremely heterogeneous group. It is therefore very difficult (if not impossible) to generalize about special education. In general, however, in the case of profound mental disabilities, targeted special pedagogy is focused on training common daily activities (skills, feeding, and toileting). With moderate or severe intellectual disabilities, the focus is on selfcare skills, employment, transportation, cooking, and social skills. In the case of a mild mental functional diversity, the chance for a full life in terms of education, employment, or social sphere is the greatest [31,32]. These individuals experience deficiencies in everyday skills, which can negatively affect their overall quality of life. To this end, there are a number of standard procedures used within the framework of comprehensive rehabilitation, which include the so-called helping professions (occupational therapy, physiotherapy, special education, psychology, social work, etc.). Video modelling and video prompting methods are also used. In general, however, the goal is to teach individuals daily life skills [8,33,34,35,36]. In view of the above, it is evident that both occupational therapy from a medical point of view and special pedagogy from an educational point of view should be fully applied to individuals with mental disabilities. The cooperation of these disciplines is important and focuses specifically on the development of the given individuals in all aspects of normal daily activities. A number of compensatory aids, standardized tests, or procedures are used for this. It is from this cooperation that the goal of our investigation arises.

The aim of this study is to apply medically used tests for assessing the functional degree of independence in children and elderly individuals with mental disabilities (WeeFIM test and FIM test) in special educational diagnostics and thus draw attention to the importance of applying these tests. The level of dependence on care that has been determined can complement the proposal of a special education plan. We present two case studies together to highlight the possibility of assessing a functional measurement of independence. We test the functional measurement of independence with versions for child clients and for adult clients. The use of these tests is limited by age. For this reason, we highlight two case studies with an age distribution such that both the WeeFIM test and the classic FIM test can be applied.

## 2. Case Description

For reasons of personal data protection, we present only basic, essential, and anonymized information without a detailed description of the individuals, for example, the relevant place of hospitalization, obtained from the analysis of medical documentation and partial interviews with the permission of legal representatives in an anonymized form. Specifically, we drew the necessary information from the translation and dismissal report with the consent and permission of the legal representatives in an anonymized form. The examination using the WeeFIM test was carried out based on the training and permission of the given device. This work also builds on our previous research investigations. We refer to these investigations in the reference list.

A boy aged 6 years and 5 months (client 1) has delayed psychomotor development, including speech development. Based on the psychological examination with the WISC III intellectual test, it was found that his overall performance was in the upper range of mild mental retardation. The verbal component of his intellect was in the range of mild to moderate mental retardation. With proper guidance and provision of a sufficient amount of stimuli in a reasonable measure, it is possible to move into the zone of light mental retardation or into the border zone. In client 1, an overall delay was noticeable for his age. Verbal expression was not developed. Monosyllabic expression in the form of interjections prevailed. Client 1 showed an effort to communicate using gestures and facial expressions. He was spontaneous, proactive, and natural in his communication. He showed limitations in his gross motor skills, especially when walking up and down stairs. He played more in parallel with other children. Except for the verbal component, he was in the border zone between mild and moderate mental functional diversity. There were uneven fluctuations between subtests. Special pedagogical examination: the examination was focused on logopaedic diagnosis. The diagnosis was focused on the assessment of four language levels. All four planes were disrupted, in this case. In the case of the pragmatic level, client 1’s unintelligible speech was noticeable. Client 1’s effort was evident, but if the other person did not understand him, the tension increased. He responded adequately to questions. The lexical–semantic level did not correspond to client 1’s age. Insufficient vocabulary, both active and passive, was evident. However, his speech understanding was not impaired. He used his own jargon, onomatopoeia, and gestures to communicate. The phonetic–phonological level was incomprehensible. He showed no interest in preschool tasks, refused to count, and managed to match colours passively. He made only brief eye contact. This was an overall slower psychomotor development. The entrance examination using the WeeFIM test gave the following results: (self-service) food: level 5—needs supervision; personal hygiene: level 3—moderate assistance; bath and shower: level 1—full assistance; dressing the upper half of the body: level 3—moderate assistance; dressing the lower half of the body: level 4—maximum assistance; toilet use: level 4—maximum assistance; control of urination and defecation: level 3—moderate assistance; (functional mobility) transfers: level 7—complete independence; transfer to the toilet: level 7—complete independence; transfer to bath: level 4—minimal assistance; walking: level 4—moderate assistance; stairs: level 5—supervision; (cognitive abilities) understanding: level 3—moderate assistance; expression: level 2—maximum assistance; social interaction: level 2—maximum assistance; problem solving: level 2—maximum assistance; memory: level 4—minimal assistance. Out of 126 possible points, the client received 66 points in testing. A girl aged 13 years and 4 months (client 2) has a combined functional diversity—hypotonia, hyperkinetic disorder with manifestations of anxiety, and unevenly distributed mental abilities in the range of moderate mental retardation. A phoniatric examination (a hearing examination) was requested at the foster mother’s request. The motivation for the request was client 2’s reduced response to speech at a distance of 4 m. The investigation was focused on the availability of otoacoustic emissions (AOE). The phoniatrist performed an auditory test—an examination of the search reaction, which showed that client 2 reacted bilaterally to weak stimulus intensity, at least from a distance of 2 to 3 m. A pro-master navigator at frequencies of 0.5–4 kHz was chosen to determine the availability of otoacoustic emissions. The resulting audiogram showed that client 2’s hearing was in the normal hearing range. A psychological examination, making use of the WISC III test, confirmed that client 2 was at the upper limit of moderate mental retardation. It also showed that a girl with a combined functional diversity requires a higher level of special educational needs. The verbal component of her intellect was in the middle zone of moderate mental retardation. Weaknesses in both fine and gross motor skills were also found, along with delayed graphomotor development and reduced visual discrimination. A special pedagogical examination showed that client 2 had slightly impaired coordination of movements in the area of gross motor skills, while in the area of fine motor skills she needed a little help when handling small objects, and in the area of tactile perception without visual support, there were significant deficiencies.

Her graphomotor expression was at the level of doodling. Her visual perception as a whole was weakened and auditory differentiation of similarly sounding words was achieved only with visual support. Her orientation in quantities of up to two was nonconstant. Her verbal expression was very poor, only in single words. Client 2 required a moderate level of special and educational measures. Postponement of compulsory school attendance was recommended. The entrance examination with the FIM test gave the following results: (personal care) food and drink: level 5—supervision; personal hygiene: level 4—minimal assistance; bath: level 7—complete independence; dressing the upper half of the body: level 4—minimal assistance; dressing the lower half of the body: level 7—complete independence; toilet use: level 7—complete independence; (sphincter control) urination and defecation: above level 7—complete independence; (movements) moving from bed, to the toilet and to the bath: after level 7—complete independence; (locomotion) walking and stairs: above level 7—complete independence; (communication) understanding: level 4—minimal assistance; expression: level 5—supervision; (social skills) social interaction: level 2—maximum assistance; problem solving: level 3—moderate assistance; memory: level 4—minimal assistance. Out of 126 possible points, the client received 101 points.

### 2.1. Summary of the Results

Based on the application of tests to assess the functional degree of independence, it was possible to quickly find out and confirm in which areas the given individual needed help. It is the division into individual subareas that gives us specific indicators of the functional level of the tested client. Eating, dressing, and communication along with social interaction appear to be more difficult. Based on these findings, we could propose the following intervention procedures.

### 2.2. Application of Special Pedagogical Intervention

Based on the examination of the boy (client 1), a special pedagogical intervention was focused on the areas of “Self-care”, “Functional mobility”, “Cognitive abilities”, and their subareas. The subareas were developed as part of a special pedagogical intervention carried out four times a day for 10 min over a period of two months. Special pedagogical intervention took place in the form of targeted individual training and was based on the determination of the zone of closest development. It was a mutual relationship between the current and potential development levels. The current level was the level client 1 was at when the examination was carried out. The potential level was the level that client 1 could reach if given support. The potential level was determined in the plan for the targeted development of the functional degree of independence on the basis of the WeeFIM test. The training of common daily activities was divided according to the individual areas of the WeeFIM test, i.e., training of breathing and grips, development of motor skills, training of dressing and undressing, preparation of clothes, development of dexterity using various games (ball, imitation, etc.) were implemented. According to the individually tested areas of the test, we also started practising common daily activities. In the case of transfers to the shower, the entry and exit from the shower corner were taught along with the use of nonslip mats and grab bars for better stability. To improve walking and dexterity, walking along a line, in a circle, or along an ellipse was applied. Additionally, walking between objects, on an inclined plane, jumping on one leg or over obstacles was included in the training. Furthermore, we specifically focused on right–left orientation using cross exercises. For better stability when walking up and down stairs, weight transfer to one leg and stepping on the curb of the stairs was practised. One- and two-step instructions related to the selected topic were given for comprehension purposes. With proper training, we achieved three-step guidelines. To help with expression, we continued the training of the correct principles of breathing (drawn from ergo therapeutic breathing gymnastics), oromotor exercises, and development of active and passive resources (reading fairy tales, naming things, commenting on activities, and matching real objects). Individual activities were also focused on the development of fine motor skills and graphomotor skills. Client 1 was supported in social interactions and educated in the principles of social behaviour.

He primarily learned greetings, expressing gratitude, and requests. To support social interaction with peers, we used board games and we also focused on enuresis and encopresis. Additionally we simulated client 1’s problem situations, e.g., turning over a plate, or breaking a plate. Client 1 was subsequently educated on how to behave appropriately in these situations, i.e., how to safely collect and throw away the shards, wash the spilt soup off himself, and ask for more soup, if necessary. The last point was memory training. In addition to the already mentioned activities that contributed to the development of memory to a greater or lesser extent, we also aimed at getting client 1 to remember people around him and everyday activities. At the same time, we used frequent feedback during the assigned activity. Figure 1 shows the input, control, and output testing. In addition, it shows the improvement in the monitored indicators.

Based on the initial FIM examination, the girl’s (client 2) level of functional independence in individual areas and subareas was determined. In the area of “Personal care”, client 2 had deficiencies in three of the six subareas (i.e., eating and drinking, personal hygiene and dressing and undressing the upper half of the body). In the areas of “Sphincter Control”, “Transfers” and “Locomotion”, client 2 had no shortcomings—she was completely independent. In the “Communication” domain, she was deficient in all subdomains (i.e., comprehension and expression). In the “Social Skills” domain, she had deficits in all subdomains (i.e., social interaction, problem solving, and memory). The special pedagogical intervention was therefore focused on the areas of “Personal care”, “Communication”, “Social skills”, and their subareas. The following table presents which subareas were developed as part of the special pedagogical intervention and also lists the activities by which the individual subareas were developed. The special pedagogical intervention took place in the form of targeted individual training in activities. It was based on the determination of the nearest development zone. In essence, it is a mutual relationship between the current and potential development levels. The current level was the level client 2 was on. The potential level was the level that a girl could reach if supported. The potential level was determined on the basis of the FIM test in the plan for the targeted development of the functional degree of independence.

With client 2, based on the results of the FIM entrance testing, we again focused on individual areas. Eating and drinking were taught in the form of training client 2 to slice food (e.g., slicing meat using compensatory aids—a special knife handle, nonslip mat), opening and closing various containers and handling them, or training in spreading butter on bread and buns and preparing some dishes. For personal hygiene, we focused on practising brushing our teeth; and for getting dressed, we focused on dressing and undressing together with the use of fine motor skills for unbuttoning and fastening buttons or zippers. In terms of communication, we focused on understanding, i.e., understanding the basic current events, e.g., on television or other media; we also trained client 2 to work with money in everyday life and its use, e.g., how much was needed to buy food. We also used model activities to train client 2 in what can take place in the shop itself. Subsequently, we trained her in orientation in the store itself. We focused on breathing and articulation exercises to help her with expression, similar to case study 1. Client 2 liked to sing, so elements of music therapy were incorporated into the training. In social skills, we instructed client 2 in social interaction. Client 2 was very communicative even with strangers, so it was necessary to prepare client 2 for communication within her personal space together with appropriate addressing (greeting) of adults and peers.

When communicating, client 2 often interrupted the speech of others, therefore she was educated in communication in everyday situations along with the basics of social behaviour. For solving problems, we applied very similar approaches as those we used with client 1, i.e., how to behave correctly in the case of spilt soup or a broken plate, and how to ask for help. Client 2 liked to travel by bus, so here we also focused on buying a ticket, the proper marking of the ticket in the means of transport and keeping it properly so that it was not lost. For memory, the focus was on remembering everyday activities. Figure 2 shows the input, control, and output testing in client 2. The graph shows the improvement in the monitored indicators.

## 3. Discussion

Intellectual functional diversity is an internationally accepted term. This term describes a deficit in cognitive and adaptive functions. The resulting deficit manifests itself in limited functioning on the intellectual level, but also on the level of adaptive behaviour and thus in limited social and practical skills [35]. In the presented case studies, we primarily focus on adaptive behaviour. Such behaviour is characterized by insufficient competence in social, conceptual, and practical skills [17]. When working with individuals with intellectual disabilities, it is necessary to realize that health status and health needs change over time and with increasing age. The changes in needs are often accompanied by a decline in vision, hearing, mobility, endurance, or selected mental processes. These risk factors must be respected and taken into account during intervention or interaction itself [37]. Individuals with this diagnosis require different levels of support. This support allows them to learn how to effectively engage in selfcare activities or how to develop intimate relationships with each other. With advancing age, the issue of employment also comes to the fore [38]. However, the barrier associated with intellectual functional diversity limits them in the learning process. It is the learning problems that bring with them the inability to sequence the individual steps in performing ordinary daily activities.

One of the possibilities of how to develop these activities is, for example, self-development learning. The essence is to practice and develop skills related to fulfilling personal hygiene, eating, but also active spending of free time outside the home environment. Typical difficulties that prevail in individuals with intellectual disabilities include taking off clothes with buttons, putting on pants, and other activities that are necessary to coordinate fine and gross motor skills [39].

In addition to model activities, more modern approaches can also be applied. These include video training or the use of Kintect V2. Augmented reality technology is coming to the forefront not only in special pedagogy but also in medical rehabilitation. Its essence is, for example, the gamification of training which consists of the child playing a video game. This video game engages them in the target behaviour, which the child then imitates in real life. Thus, the positive impact of video games on the development of common daily activities can be used [40,41]. A number of technologies are used in the field of special education to improve the overall outcomes of students with disabilities. One such approach is the already mentioned video modelling. This is a teaching method that instructs students to watch a short video. This video shows some activities. They then imitate one of these activities in real time. The same principle is then applied to video prompts that encourage the student to perform a certain activity [42,43].

In addition to those mentioned above, there are a number of approaches, whether conventional or unconventional, or from an educational or medical point of view allow the training and development of everyday activities. In general, activity limitation is an important dimension of functional diversity. Despite this, instruments that measure the limitation of activities in individuals with developmental disabilities, in our case with intellectual functional diversity, are not fully widespread. Such tests primarily target people with dementia, or after surgery, stroke, or lung disease, etc. [44,45,46]. In general, children with mental retardation are children with special needs and with an intelligence level >70 IQ. The self-concept of such these individuals is relatively strongly influenced by the pattern of care and the environment [47]. Educational activities or targeted therapies are the main areas of support. This support leads to the development of compensation mechanisms. Compensation mechanisms can be effective in the event that educational activities are based on a methodical approach that reflects the particularity of the individual’s development. An approach that focuses on structuring behaviour is also needed to manage normal daily activities. This approach includes supporting cognitive function, language, and socialization. For example, language skills are acquired better and easier in conversational interactions in certain contexts. Initially, the child learns to plan time periods for structured social interactions, which also leads to the improvement of coping with common daily activities [36]. The essence of this article is primarily to enrich special education practice. It is not customary for a special education educator to assess functional measures of independence. Through this study, we want to show the importance of this assessment, which can be performed by the special educator themself and can evaluate the current state of the given client in real time with objective results.

The evaluation using the standardized tests mentioned was crucial for us. We know that other tests can be used to assess cognitive performance, attention, and other areas. Our goal was to apply tests that primarily focus on assessing the mastery of everyday activities. It is this kind of assessment that we consider important if we want to improve and, above all, promote social adaptability in the individual. In special education, the trend is more towards tests that assess school performance or cognitive function itself. Consequently, the area of social adaptability is neglected which is why we have taken this area into account in the preparation of this study. We see this area as pivotal and aim to contribute to the practice of special education precisely by observing and assessing how clients cope with daily living activities.

## 4. Conclusions

The use of tests aimed at evaluating the functional degree of independence appears to be useful to complement other test methods in special pedagogy. In addition to the evaluation of motor or cognitive levels, an important area of interest is also the focus on managing ordinary daily activities. It follows from the text that the learning process is to some extent limited by the degree of mental functional diversity. This is reflected not only in communication or social interaction, but also in living an independent life. This impairs the perception of the very quality of life. Complex diagnostics can thus create a test background that provides an overall view of the given individual and points to the direction in which the intervention itself should go. We are aware that the mentioned tests are primarily used in treatment and rehabilitation fields, where they undoubtedly have a strong presence. In special pedagogy, even though its intervention specifically develops normal daily activities, it is not customary to test these areas. However, from a long-term perspective, we observe a very close connection between special pedagogy and occupational therapy. Their intervention procedures often overlap [8]. It is the cooperation of these fields in practice that is very beneficial not only for the client themself but also for their family and surroundings. In the same way, we perceive this form of testing the cognitive functions, but also precisely in the evaluation of the functional degree of independence, to be beneficial. Based on these conclusions, the following questions arise:

In the event that several experts have an influence on a given individual, would each expert carry out testing with his own test? Or, is it possible to apply the results from one test and based on this adapt the intervention of special pedagogy, occupational therapy, physiotherapy, etc.?Is it possible to use the given test immediately, without knowledge and experience?

In response to the questions, it is necessary to mention that each branch of the so-called field of helping professions treats the given individual using its own approaches. Experts use a number of compensatory tools and educational procedures, etc. From our point of view, an effective situation appears to be where each participating expert would performs their own testing.

## Figures and Tables

**Figure 1 ijerph-19-15474-f001:**
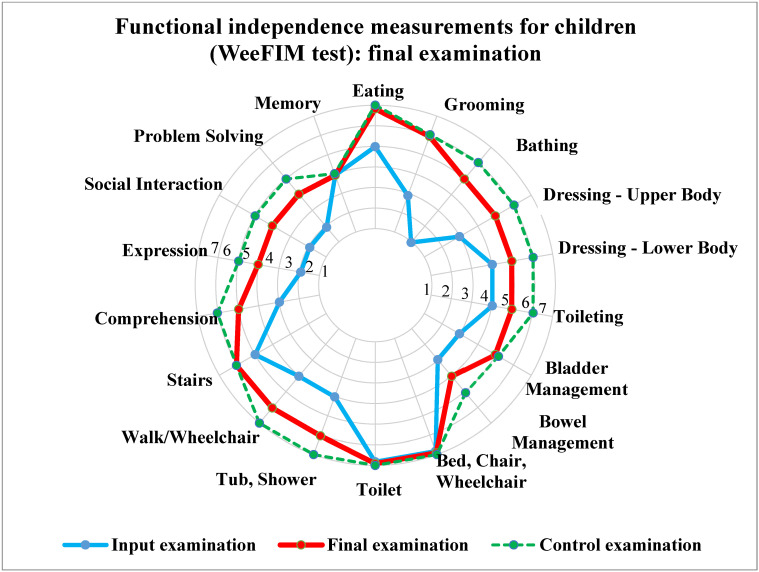
Results of the WeeFIM test for a boy (client 1) aged 6.5 years.

**Figure 2 ijerph-19-15474-f002:**
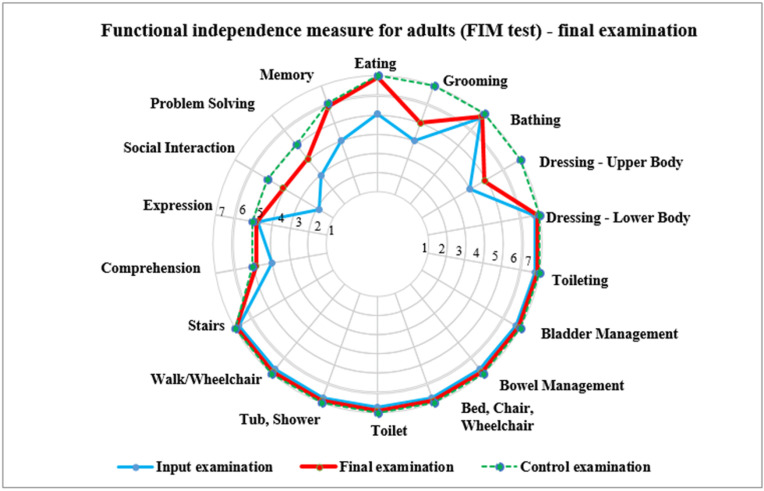
Results of the FIM test for a girl (client 2) aged 13.4 years.

## Data Availability

The data are available upon reasonable request.

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
