# Peer review of "Assessment of the Functional Level of Independence in Individuals with Mental Disabilities as Part of Special Education Diagnostics: Case Studies"

_ijerph, 2022, doi:10.3390/ijerph192315474_

Round 1

Reviewer 1 Report

The Authors submitted a paper about the use of functional indipendence tests in special pedagogy.

In the Introduction The first two references are not reachable on Pubmed, and the first one does not estimate the prevalence of Intellectual disability in the general population. Where are these numbers from?

Reference number 6, 8 are a self citation from Lankova B. and should be acknowledged as such in the text of the paper at least.

Reference number 8 is a self citation fom Vostry Mand should be acknowledged as such in the text of the paper at least.

Reference number 8 is a self citation from Pesatova I. and should be acknowledged as such in the text of the paper at least.

In the Introduction is mentioned a difference in the medical model and the social model of disability, hinting that functional limitation is different in them. this should be expanded because it's not clear in the text.

Where is reference number 4 and 16 are the data about severe intellectual disability having also severe motor disabilities?

Section 1.1 should have a reference about the definition of mental disability and all the skills mentioned.

"These skills are needed to manage normal daily activities (also daily activities)" What is the meaning of this sentence?

" They require constant supervision and, together with maladaptive behaviour and self-harm, this is a serious problem in the care of these individuals" This sentence needs a reference.

"Limits in gross motor skills, especially when walking up and down stairs." This sentence needs a verb.

Graph 1 and Graph 2 are the same image; that directly contradicts the text.

41 and 42 are the same reference

Author Response

Thank you for your review, which I have incorporated. 

1. 3% was an error, it is 1% in the population. This information is given by source 2.

2. Autocorrelations are also entered in the text and highlighted

3. The Vostry association is also included. Thank you for your attention, unfortunately the article is a continuation of the previous investigation, so I refer to it.

4. The references are highlighted in the text

5. Skills are understood as activities of daily living that the individual performs either independently or with assistance. It depends on the degree of disability.

The 6th chart is corrected, thank you. We overlooked and inserted the same chart by mistake.

7. Links corrected.

Thank you again for the accurate check. We appreciate it.

Translated with www.DeepL.com/Translator (free version)

Reviewer 2 Report

The Manuscript: „ Assessment of the functional level of independence in individuals with mental disabilities as part of special education diagnostics - Case studies’’ by Michal Vostrý and colleagues  analyse the application of standardized tests in evaluating the functional degree of independence in children (WeeFIM test). Based on the outcomes of the study, the authors further discuss the suitability of using functional independence tests in practices. After going through the manuscript, I have few comments for the authors:

1.     In the abstract, it is mentioned that the study focuses on functional degree of independence in children (WeeFIM test) and adults (FIM test). However, both the reported cases are children. Please correct this discrepancy.

2.     The Introduction part is messy and too long. Please shorten it.

3.     Figures 1 and 2 with results of the WeeFIM test need to be better explained with exact procedures of input, control and final examinations.

4.     Conclusion is way too long. Please trim it so that a concise and to the point message of the study could be conveyed.

5.     There are numerous grammatical and syntax errors in the manuscript. I would suggest double checking of the manuscript to minimize grammatical flaws.

Author Response

Dear reviewer,

Thank you for your review and we have incorporated all comments.

1. abstract corrected. It is the age range of the test in question, although the WeeFIM test is for a child up to 7 years old and the FIM test is for older children and this corresponds to client 2.

2. remove a paragraph that is already repetitive and thus reduce the introductory part.

3. We will also reduce the conclusion by a paragraph, so the focus will be on the key information

4. the text has been proofread by a native speaker, thank you and apologies for the errors.

Once again, thank you very much for your time and your helpful review, which helped to improve the article.  Dear Reviewer. Unfortunately it was not possible to shorten the introduction in a way that would probably suit you fully. The reason is a different opinion of another reviewer who would have added some information. Therefore, we have removed the general paragraph, which is not so essential and does not change the quality of the introduction itself.

Reviewer 3 Report

Dear Authors, I read your work “Assessment of the functional level of independence in individuals with mental disabilities as part of special education diagnostics - Case studies” and here I enclose my recommendations to you:

General Comments

1.     I suggest the Authors to have their text edited for English Language.

2.     The aim of the study is clear, and it is expected to see an RCT study or a Cohort study, but we see case studies, how this is supported.

Introduction

The introduction is sound and has a good rational behind it but sure there is space for improvement.

Case Study

The two-case studies have sufficient information, but the profiling is not so good. The Author’s state in this part “…Special pedagogical examination: the examination was focused on logopedic diagnosis….” With this the Authors give to the reader an information that the evaluation of special education needs is the same as the logopedic one. It is different to administrate SLT assessments in order to evaluate the language speech and communication profile or a “client” and it is different to copy a “client” needs for special education. I suggest the Authors to re-write this, since I believe is a mistake. Additionally, we do not see what assessment were followed for both pedagogical and logopedic evaluations and I strongly suggest the Authors to provide this information.

Furthermore, when a case study is presented it is expected to be a unique case. In those two cases what makes them unique, besides using the system that is presented? I suggest the Authors to underline this in their text.  

Discussion & Conclusion

The discussion and conclusion had a good structure. It is though unclear to the reader what is the adding value of this work. It was expected from the title, the abstract and the two cases to read something new that would add data-information-knowledge to this scientific field. I suggest the Authors to consider keep the structure and change what they state in this section.

Thank you.

Author Response

Dear Reviewer,

Thank you for your review, which we appreciate. We agree with your points. We have incorporated these points and added them to the text. The point was to point out the appropriateness of using these standardized tests in the practice of the special educator. Since this rule is not customary. If we start from the essence of special education approaches, we are not only interested in the degree of educational attainment in practice, but we are also interested in promoting social adaptability.

1. The text was reviewed by a native speaker for proofreading.

2. We present a mixed case study of different probands, where we focus on two tests aimed at coping with activities of daily living, the WeeFIM and the FIM test. We point out their importance in special education diagnosis, as the main tests that are primarily used in medical rehabilitation. However, tests on this topic are neglected in educational rehabilitation. We see their potential primarily in the individuality of the individuals in question and also in the fulfilment of the goals of special education. The present article is thus conceived in this direction.

Reviewer 4 Report

Summary: It is recommended that it be structured according to each section of the article: introduction, objectives, methodology, results and main conclusions.

Introduction:

- Authors are recommended to use the term functional diversity, not the term disability.

- In order to justify the relevance of addressing the issue raised, authors are recommended to carry out a more exhaustive review of the research that precedes the interest in addressing their research. In this sense, it is recommended that they review studies that address the importance of researching this topic in recent years aimed at improving the inclusion and educational response to this group. For example, the following:

-Villa-Fernández, N., & Martín-Gutiérrez, Á. (2020). Educación inclusiva y digital: desafíos y propuestas a partir del COVID-19. Virtu@ lmente, 8(2), 7-27. https://doi.org/10.21158/2357514x.v8.n2.2020.2715

-Mapes, B. M., Foster, C. S., Kusnoor, S. V., Epelbaum, M. I., AuYoung, M., Jenkins, G., ... & All of Us Research Program. (2020). Diversity and inclusion for the All of Us research program: A scoping review. PloS one, 15(7), e0234962.

-Chakravarthi, B. R., & Muralidaran, V. (2021, April). Findings of the shared task on hope speech detection for equality, diversity, and inclusion. In Proceedings of the first workshop on language technology for equality, diversity and inclusion (pp. 61-72).

Methodology:

- It is important to clearly state the objectives of the research. Authors are recommended to clearly state the general and specific research objective in the methodological section.

- The methodology should be clearly described in a section, as well as the triangulation of the results. It is important to present the results on a quantitative level with the spider graphs that are presented, but also on a qualitative level with the assessments made by the people who take the standardized tests. This last part enriches the results presented. 

- The sample selected is very small, so it is advisable to clarify in detail the selection criteria used. It is not advisable to indicate the average age of the participants, but rather the real or average age without decimals.

- We would like to thank the authors for detailing in the methodology section the standardized instruments used for each of the dimensions selected in this study.

- In the statistical analysis used, it is advisable not only to perform descriptive but also a binary logistic regression with the aim of presenting the relationship between some selected variables.

Results:

- It is recommended to present the results section because of the specific objectives pursued by the research. As the results are, at the moment, they are not clear to the reader. In this sense, it is recommended that the specific objectives of the work be redefined.

- The figures presented should follow the citation rules of the journal. In addition, they should be smaller (they are too large) and centered on the page.

-Discussion, conclusions, limitations, and outlook:

- Authors are recommended to present the last sections of the article in the following order: Discussion, conclusions and theoretical and practical implications, limitations and prospective:

- At this point the discussion is insufficient, it should be better argued and presented by the specific objectives of the research.

 Citations and references and regulations:

- Authors should review the presentation of tables and figures in accordance with the guidelines established in the journal.

- It is important to review the citations and bibliographical references in the full text, in accordance with the journal's citation regulations. In addition, there are citations that end with "," instead of ".".

- It is recommended to incorporate some bibliographical references that could be of interest to focus on the relevance of this work at the present time.

 Ethics of the contribution: The authors have indicated their ethical commitment, but it would be important to indicate the ethical code of authorization granted by the university.

Author Response

Dear Reviewer,

Thank you for your review.

  1. We have tried to incorporate everything into the article to meet the required quality.
  2. In the case of age, we do not give an average age, but it is an abbreviation of 6 years and 5 months, i.e. 6.5 years.
  3. We have added the quote you recommended, thank you for adding it.

4.We have also added better conclusions and wording to the results.

We appreciate your feedback, which contributed to the improvement of the article. Hopefully we have managed to incorporate everything properly.

Thank you again.

Reviewer 5 Report

The study is uniquely clear and has the quality to contribute to the literature. I expect the authors of the article to clarify three issues. What was done in the Intervation programs? It needs to be explained briefly. Issues such as validity and reliability, pilot study need to be addressed.

Information about the procedure of the study should be given. The questions of how the data were collected and how the data were analyzed should be addressed under a separate heading in the article.

The limitations of the research should be explained in detail and should be arranged under a separate title before the results section.

Author Response

Dear Reviewer,

  1. We have tried to incorporate everything into the article to meet the required quality.
  2. We appreciate your review and the objective errors we have corrected.
  3. The data were obtained in the framework of our survey, which we refer to in the text. Thank you for these comments.

Round 2

Reviewer 1 Report

I believe the Authors have responded to most of my points.

"Limits in gross motor skills, especially when walking up and down stairs" this sentence still needs a verb, like "Limits are present in..." or "There are limits in..."

The first graph in page 9 is the same as Graph 1

The self-references should be acknowlegded in the text.

Author Response

Dear Reviewer,

thank you for your feedback

1. we have added the verb to the text
2. the graph shows similar values, but they are two different ones. The first graph is on the results of the WeeFIM test and the second is on the FIM test.
3. we have acknowledged the auto-reference.

Reviewer 2 Report

I am happy with authors' response to my comments.

Author Response

Dear Reviewer,

we are happy that you are satisfied. Thank you once again for your professional opinion. which has enabled us to present the results and the paper better. We appreciate it.

Reviewer 3 Report

Dear Authors, 

I read your revised work and I have one suggestion to you: Please, make more sound you evaluation methods for both case studies. The readers must understand and know which were the assessments used for SLT difficulties, for Special Education difficulties and Social difficulties. It will make sounder your methods and clarify more the what was the exact profile of those case studies. Additionally, it will make crystal clear how all the reported scores of all assessments will can contribute to each case and their level of independence. 

Thank you. 

Author Response

Dear Reviewer,

thank you for your feedback. 

1. we have added the necessary information to the discussion to reflect your request. 
Thank you again. Your review helped us to better prepare the article and to focus on the main results.